# Unbiased identification of cell identity in dense mixed neural cultures

**Sarah De Beuckeleer[1], Tim Van De Looverbosch[1], Johanna Van Den Daele[1], Peter Ponsaerts[2], Winnok H De Vos[1,3,4]***

[1]Laboratory of Cell Biology and Histology, University of Antwerp, Antwerp, Belgium; [2]Laboratory of Experimental Haematology, Vaccine and Infectious Disease Institute (Vaxinfectio), University of Antwerp, Antwerp, Belgium; [3]Antwerp Centre for Advanced Microscopy, University of Antwerp, Antwerp, Belgium; [4]µNeuro Research Centre of Excellence, University of Antwerp, Antwerp, Belgium

## eLife Assessment

This study presents an **important** application of high-content image-based morphological profiling to quantitatively and systematically characterize induced pluripotent stem cell-derived mixed neural cultures cell type compositions. **Exceptional** evidence through rigorous experimental and computational validations support new potential applications of this cheap and simple assay.

**\*For correspondence:**
winnok.devos@uantwerpen.be

**Abstract** Induced pluripotent stem cell (iPSC) technology is revolutionizing cell biology. However, the variability between individual iPSC lines and the lack of efficient technology to comprehensively characterize iPSC-derived cell types hinder its adoption in routine preclinical screening settings. To facilitate the validation of iPSC-derived cell culture composition, we have implemented an imaging assay based on cell painting and convolutional neural networks to recognize cell types in dense and mixed cultures with high fidelity. We have benchmarked our approach using pure and mixed cultures of neuroblastoma and astrocytoma cell lines and attained a classification accuracy above 96%. Through iterative data erosion, we found that inputs containing the nuclear region of interest and its close environment, allow achieving equally high classification accuracy as inputs containing the whole cell for semi-confluent cultures and preserved prediction accuracy even in very dense cultures. We then applied this regionally restricted cell profiling approach to evaluate the differentiation status of iPSC-derived neural cultures, by determining the ratio of postmitotic neurons and neural progenitors. We found that the cell-based prediction significantly outperformed an approach in which the population-level time in culture was used as a classification criterion (96% vs 86%, respectively). In mixed iPSC-derived neuronal cultures, microglia could be unequivocally discriminated from neurons, regardless of their reactivity state, and a tiered strategy allowed for further distinguishing activated from non-activated cell states, albeit with lower accuracy. Thus, morphological single-cell profiling provides a means to quantify cell composition in complex mixed neural cultures and holds promise for use in the quality control of iPSC-derived cell culture models.

## Introduction

Modelling the complexity of the human brain and its (dys)function has proven to be notoriously challenging. This is due to its intricate wiring, the cellular heterogeneity, and species-specific differences (*De Strooper and Karran, 2016*). To increase the translational value of neuroscientific research, there is a need for physiologically relevant human models. With the advent of human iPSC technology, it has become possible to generate a wealth of brain-resident cell types including neurons (*Bell*

et al., 2019), astrocytes (*Neyrinck et al., 2021*), microglia (*Haenseler et al., 2017*), oligodendrocytes (*Nevin et al., 2017*), and endothelial cells (*Abutaleb and Truskey, 2021*), allowing the study of complex polygenic pathologies that cannot be modelled in animals, opening avenues to precision pharmacology (*Paik et al., 2020*; *Lopez-Gonzalez et al., 2016*). Furthermore, the ability of iPSC to develop into organoids renders them attractive for studying multi-cellular interactions in a 3D context that is closer to the in vivo situation (*Amin and Paşca, 2018*). However, genetic drift, clonal, and patient heterogeneity cause variability in reprogramming and differentiation efficiency (*Rouhani et al., 2014*; *Salomonis et al., 2016*). The differentiation outcome is further strongly influenced by variations in protocol (*Yamamoto et al., 2022*). Together, it leads to inconsistent and potentially misleading results and consequently, it hinders the use of iPSC-derived cell systems in systematic drug screening or cell therapy pipelines. This is particularly true for iPSC-derived neural cultures, as their composition, purity, and maturity directly affect gene expression and functional activity, which is essential for modelling neurological conditions (*Kuijlaars et al., 2016*; *Hernández et al., 2022*). Thus, from a preclinical perspective, there is a need for a fast and cost-effective quality control (QC) approach to increase experimental reproducibility and cell type specificity (*D'Antonio et al., 2017*). From a clinical perspective, in turn, robust QC is required for safety and regulatory compliance (e.g. for cell therapy). To address this lack of standardisation, large-scale collaborative efforts have been set up such as the International Stem Cell Banking Initiative (*Sullivan et al., 2018*), which focusses on clinical quality attributes and provides recommendations for iPSC validation testing for use as cellular therapeutics, or the CorEuStem network, aiming to harmonize iPSC practices across core facilities in Europe. Current culture validation methods include combinations of sequencing, flow cytometry and immunocytochemistry (*D'Antonio et al., 2017*; *Chen et al., 2021*). These methods are often low in throughput, costly and/or, destructive. Hence, we set out to develop a method for evaluating the composition of such cultures based on high-content imaging, which is fast, affordable, and scalable. The primary goal was to facilitate cell type identification in dense cultures, while ensuring compatibility with subsequent immunocytochemistry or molecular assays for further biological inquiries. To this end, we employed the cell painting (CP) (*Gustafsdottir et al., 2013*; *Bray et al., 2016*) method, which is based on labelling cells with simple organic dyes and analysing the resulting phenotypes. CP has proven to be a powerful and generic method for predicting the mode-of-action of pharmacological or genetic perturbations, and this sheerly using a cell morphology readout (*Cimini et al., 2023*; *Way et al., 2021*; *Schiff et al., 2022*). Thus far, CP has primarily been used to predict conditions associated with pharmacological treatments or genetic modifications using images as input. This way, CP has allowed predicting patient diversity in lung adenocarcinoma-associated somatic variants *Caicedo et al., 2022* or genetic variation across donors in iPSC cultures (*Tegtmeyer et al., 2024*). Methods such as PhenoRipper *Rajaram et al., 2012* and CP-CHARM (*Uhlmann et al., 2016*) use whole-image features for classification, which circumvents the difficulty of cell segmentation and allows the classification of images with cells of similar phenotype or class (*Caicedo et al., 2017*). However, they do not consider differences in cell density and disregard the inter-cellular heterogeneity within the field of view. Therefore, we explored the amenability of CP to predict individual cell types in dense and mixed cultures. By combining deep learning for cell segmentation and classification, we established an approach that allows recognizing of cell types with high accuracy, even in very dense cultures. We employed the approach to evaluate the composition of iPSC-derived neural cultures as they are often quite dense and composed of heterogenous cell types. Varying the density, composition, and cell state allowed us to benchmark the discriminatory potential of cell-based profiling.

## Results
### Neural cell lines have a unique morphotextural fingerprint

Several groups have demonstrated that morphological cell profiling can be used to discriminate pharmacogenomic perturbations based on phenotypic similarity (*Cimini et al., 2023*; *Way et al., 2021*; *Schiff et al., 2022*). We asked whether a similar approach could be leveraged to unequivocally distinguish individual cell types as well. To this end, we implemented a version of CP based on 4-channel confocal imaging (*Figure 1A*) and first applied it to monocultures of two neural cell lines from a different lineage, namely astrocyte-derived 1321N1 astrocytoma cells and neural crest-derived SH-SY5Y neuroblastoma cells. First, we explored whether traditional morphotextural feature

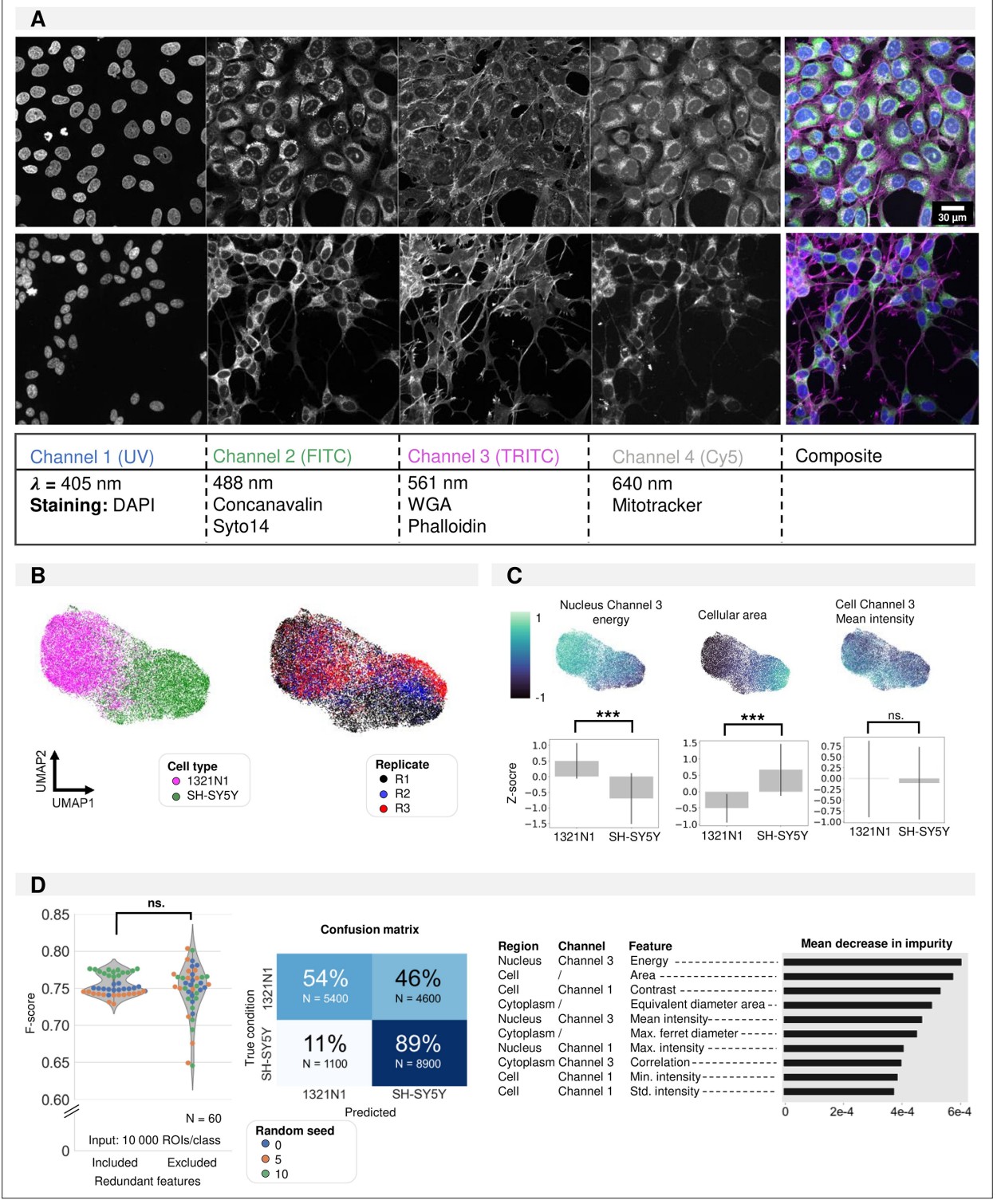

**Figure 1.** Shallow classification using manually defined features. (**A**) Image overview of 1321N1 (top) and SH-SY5Y (bottom) cells after cell painting (CP) staining acquired at higher magnification (Plan Apo VC 60xA WI DIC N2, NA = 1.20). Scale bar is 30 µm. The channel, wavelength, and dye combination used is listed in the table below the figure. This color code and channel numbering is used consistently across all figures. (**B**) UMAP dimensionality reduction using handcrafted features. Each dot represents a single cell. The colour reflects either cell type condition (left) or replicate (right). This shows that UMAP clustering is a result of cell type differences and not variability between replicates. (**C**) Feature importance deducted from the UMAP (feature maps). Each dot represents a single cell. Three exemplary feature maps are highlighted alongside the quantification per cell type. These feature representations help understand the morphological features that underlie the cluster separation in UMAP. (**D**) Random Forest classification performance

*Figure 1 continued on next page*

*Figure 1 continued*

on the manually defined feature data frame with and without exclusion of redundant features. Average confusion matrix (with redundant features) and Mean Decrease in Impurity (reflecting how often this feature is used in decision tree splits across all random forest trees). All features used in the UMAP are used for random forest (RF) building. Each dot in the violinplot represents the F-score of one classifier (model initialisation, N=30). Classifiers were trained 10 x with three different random seeds.

The online version of this article includes the following figure supplement(s) for figure 1:

**Figure supplement 1.** PCA dimensionality reduction on handcrafted features extracted from monocultures of 1321N1 and SH-SY5Y cells.

extraction provided sufficient distinctive power. The features were calculated for each channel in three regions of interest, namely the nucleus, cytoplasm, and whole cell. They describe the shape, intensity, and texture features of each ROI (*Table 1*). Representation of the resulting standardized feature set in UMAP space revealed a clear separation of both cell types along the first UMAP dimension. Clustering of instances was less pronounced after principal component analysis (*Figure 1—figure supplement 1*), but UMAP better preserves local and global data structure (*McInnes et al., 2018*). Despite some separation of replicates across the second UMAP dimension, the absence of clear replicate clusters

**Table 1.** Definition of handcrafted features (according to the scikit-image documentation).

All of these features were extracted for each fluorescent channel (1-4) and region (nucleus, cytoplasm, and whole-cell) in the cell painting images.

| Feature name | Description | Category |
| --- | --- | --- |
| Intensity max | Maximal pixel intensity inside the ROI | Intensity |
| Intensity mean | Mean pixel intensity inside the ROI | Intensity |
| Intensity min | Minimal pixel intensity inside the ROI | Intensity |
| Intensity Std | Standard deviation of the pixel intensities inside the ROI | Intensity |
| Area | Area of the ROI | Shape |
| Area convex | The area of the convex polygon that encloses the ROI | Shape |
| Area filled | Area of the ROI including all holes inside the ROI | Shape |
| Axis major length | The length of the major axis of the ellipse that has the same normalized second central moments as the region. | Shape |
| Axis minor length | The length of the minor axis of the ellipse that has the same normalized second central moments as the region. | Shape |
| Centroid | Coordinates of the center of the ROI | Shape |
| Eccentricity | Eccentricity of the ellipse that has the same second-moments as the ROI. | Shape |
| Equivalent diameter area | The diameter of a circle with the same area as the ROI. | Shape |
| Extent | The ratio of the Area of the ROI to the Area of the bounding box around the ROI. | Shape |
| Feret diameter max | The longest distance between points around a ROIs' convex polygon. | Shape |
| Orientation | The angle between the X-axis and the major axis. | Shape |
| Perimeter | Length of the contour of the ROI | Shape |
| Perimeter crofton | Perimeter of the ROI approximated by the Crofton formula in four directions. | Shape |
| Solidity | The ratio of the number of pixels inside the ROI to the number of pixels of the convex polygon. | Shape |
| Contrast | The difference between the maximum intensity and minimum intensity pixel inside the ROI. | Texture |
| Dissimilarity | The average difference in pixel intensity between neighbouring pixels. | Texture |
| Homogeneity | Value for similarity of pixels in the ROI. | Texture |
| Energy | Value for the local change of pixel intensities in the ROI. | Texture |
| Correlation | Value for the linear dependency of pixels in the ROI. | Texture |
| ASM (angular second moment) | Value for the uniformity of pixel values in a ROI. | Texture |

showed that the morphological differences between cell types were consistent across biological replicates (*Figure 1B*). When projecting individual features onto the UMAP space, we found that both texture (e.g. Nucleus Channel 3 Energy) and shape (e.g. Cellular Area) metrics contributed to the cell type separation (*Figure 1C*). The contribution of intensity-related features (e.g. Channel 3 Intensity) to cell type separation was less pronounced as they were more correlated with the biological replicate. Thus, we conclude that cell types can be separated across replicates based on a morphotextural fingerprint.

## Convolutional neural network outperforms random forest in cell type classification

To evaluate whether the morphotextural fingerprint could be used to predict cell type, we performed Random Forest (RF) classification using the full feature dataset, using different seeds for splitting up the data into training and validation sets. This resulted in a rather poor accuracy (F-score: 0.75±0.01), mainly caused by the significant (46%) misclassification of 1321N1 cells (*Figure 1D*). The imbalance in recall and precision was surprising given the clear separation of cell populations in UMAP space using the same feature matrix. When inspecting the main contributions to the RF classifier using the mean decrease in impurity, we found very similar features as highlighted in UMAP space to add to the discrimination (*Figure 1C*). Where the most important features (e.g. Cellular Area, Channel 1 Contrast) showed the expected gradient along the first UMAP direction, lower ranked parameters (e.g. Cellular Channel 3 Mean Intensity) had no contribution to UMAP separation (*Figure 1C*). Reducing noise by removing redundant features (correlation >0.95) could not ameliorate RF classification performance, which may be due to the documented bias in feature selection for node splitting in high-dimensional data (*Nguyen et al., 2015*). This result drove us to evaluate a different classification approach based on a ResNet (*He et al., 2015*) convolutional neuronal network (CNN). Here, we no longer relied on the extraction of 'hand-crafted' features from segmented cell objects for training the shallow RF classifier. Instead, we used isotropic image crops of 60 µm centred on individual cell centroids and blanked for their surroundings as input for the CNN (*Figure 2A*). Using this approach, we found a significantly higher prediction performance (F-score of 0.96±0.01), with a much more balanced recall and precision (*Figure 2B*). The UMAP space built from the CNN feature embeddings showed a clear cell type separation (*Figure 2—figure supplement 1A*). Even with only 100 training instances per class, the CNN outperformed RF, but optimal performance was attained with 5000 training instances (*Figure 2C*). Both RF and CNN models trained on a combination of three biological replicates ('Mixed reps.') performed similarly as models that were trained on only a single replicate ('Single reps.') (*Figure 2D*). However, a model trained on a dataset containing multiple replicates ('Mixed reps.') outperformed CNN models that were only trained on a dataset with less variability ('Single rep.') when predicting instances of an independent unseen replicate (*Figure 2E*). This emphasises the need for including sufficient variation in the training set. Although much more performant, CNN classification does not allow direct retrieval of intuitive features, which complicates model interpretation. To gain a visual understanding of image information contributing most to the classification we resorted to Gradient-weighted Class Activation Mapping (Grad-CAM)(*Selvaraju et al., 2020*). This revealed that the attention of the CNN was mainly focused on cell borders (edge information), and nuclear and nucleolar signals (*Figure 2F*). When scrutinizing CNN misclassifications, we found that these are mainly caused by faulty cell detection (e.g. oversegmentation of cell ramifications), unhealthy/dead cells, debris or visibly aberrant cell shapes (*Figure 2—figure supplement 1B*) - errors not necessarily attributed to the CNN.

To shed light on the contribution of individual markers to the classification, we eroded the input to single-channel images. For all cases (each individual channel), the prediction performance was below or equal to 85,0% indicating that no single channel contains all relevant information (*Figure 3A*). Combinations of two or three channels could not match the prediction accuracy of the full four-channel image either. Thus, we concluded that all channels contribute to the successful classification. As the CNN directly uses image crops as input, the image quality will determine the classification performance. Therefore, we assessed the impact of resolution and signal-to-noise ratio (SNR). We simulated the effect of decreasing spatial resolution through progressive pixel binning from 192 pixels (original, pixel size 0.3 µm) to 9 pixels (pixel size 6 µm). For each iteration, three CNN models were trained and evaluated. A reduction in pixel size from 0.3 µm to 0.6 µm did not result in a significantly lower prediction performance, but decreasing the spatial resolution further caused a progressive decrease

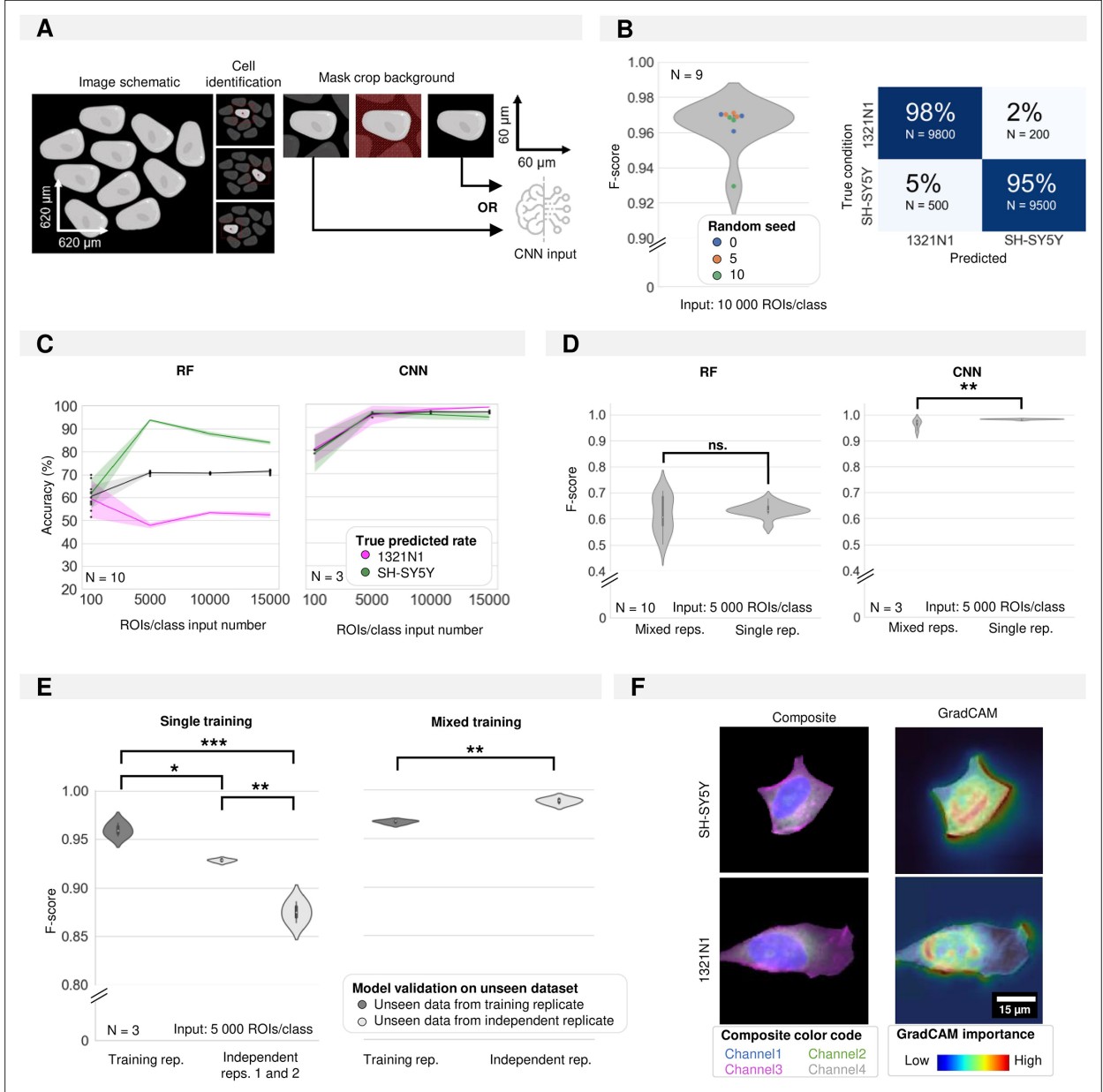

**Figure 2.** Convolutional neural network classification of monoculture conditions. (**A**) Schematic of image pre-processing for convolutional neuronal network (CNN) classification. (**B**) CNN accuracy and confusion matrix on cell type classification in monocultures. Each dot in the violinplot represents the F-score of one classifier (model initialisations, N=9). (**C**) Shallow vs. deep learning accuracy for a varying number of input instances (region of interests, ROIs). Each dot in the violinplot represents the F-score of one classifier (model initialisation, **N**). The ribbon represents the standard deviation on the classifiers. (**D**) The impact of experimental variability in the training dataset on model performance for shallow vs. deep learning. Classifiers were trained on either 1 (single rep.) or multiple (mixed reps.) replicates (where each replicate consists of a new biological experiment). Mann-Whitney-U-test, p-values resp. 0.7304 and 0.000197. Each violinplot represents the F-score of N classifiers (model initialisations). (**E**) The performance of deep learning classifiers trained in panel D on single replicates (low variability) or mixed replicates (high variability) on unseen images from either the training replicate (cross-validation) or an independent replicate (independent testing) (where each replicate consists of a new biological experiment). Kruskal-Wallis test on single training condition, p-value of 0.026 with post-hoc Tukey test. Mann-Whitney-U-test on mixed training, p-value of 7.47e-6. Each violinplot represents the F-score of one classifier (model initialisation, N=3). (**F**) Images of example inputs given to the CNN. The composite image contains an overlay of all cell painting (CP) channels (left). The GradCAM image overlays the GradCAM heatmap on top of the composite image, highlighting the most important regions according to the CNN (right). One example is given per cell type.

The online version of this article includes the following figure supplement(s) for figure 2:

**Figure supplement 1.** Cell classification in monoculture conditions.

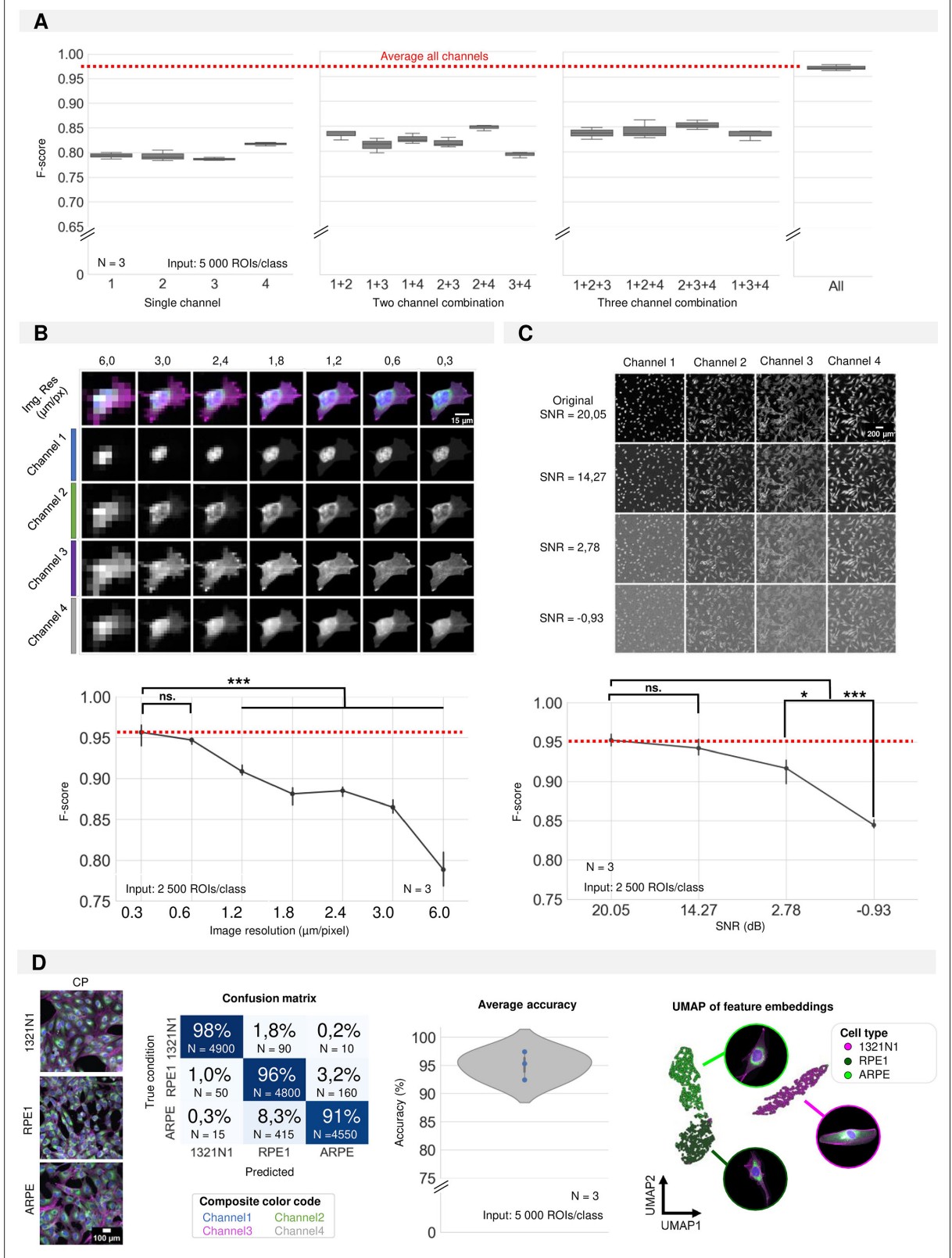

**Figure 3.** Required input quality for cell type classification. (**A**) Performance of the convolutional neuronal network (CNN) when only a selection of image channels is given as training input. Each boxplot represents the F-score of one classifier (model initialisation, N=3). Channel numbering is in accordance with *Figure 1A*. (1=DAPI; 2=FITC; 3=Cy3; 4=Cy5) (**B**) Simulation of the effect of increased pixel size (reduced spatial resolution) on the classification performance. Each point is the average of three CNN model initializations (**N**), with bars indicating the standard deviation between

*Figure 3 continued on next page*

*Figure 3 continued*

models. Red line indicates the average F-score of the original crops. (**C**) Simulation of the effect of added Gaussian noise (reduced signal-to-noise ratio) on the classification performance. Each point is the average of three CNN model initialisations, with bars indicating the standard deviation between models. Red line indicates the average F-score of the original images. Statistics were performed using a Kruskal-Wallis test with post-hoc Tukey test. (**D**) CNN performance on 1321N1, RPE1, and ARPE cells. Dots represent different model initialisations (N=3).

The online version of this article includes the following figure supplement(s) for figure 3:

**Figure supplement 1.** Influence of the nuclear/cellular size on convolutional neuronal network (CNN) prediction and association with the input background (space artificially set to zero outside of the segmentation mask).

in F-score (*Figure 3B*). In a similar manner, we tested the impact of decreasing SNR (by increasing the level of Gaussian noise) on classification accuracy. Starting from an original SNR value of 20.05 dB, we found that the F-score started to decrease when lowering the SNR below 14.27 dB (*Figure 3C*).

We observed that even with low input image quality, the CNN performance retained approximately 80% accuracy. We attribute this to the predictive power of the nuclear size, which is a dominant discriminating feature between the cells (*Figure 3—figure supplement 1*). Given the relatively overt phenotypic difference between 1321N1 and SH-SY5Y cells, we asked whether the CNN could also discriminate more subtle phenotypes. To this end, we used 1321N1 cells and two related retinal pigment epithelial cell lines, hTERT-immortalized RPE1 and the spontaneously immortalized variant ARPE (*Figure 3D*). The CNN discriminated all three cell lines with a high accuracy of 95,06±2,51%. Unsupervised UMAP dimensionality reduction using the CNN feature embeddings revealed three clusters of ROIs, with the two RPE lines located closer together - but still well separated - in comparison to the morphologically more distinct 1321N1 cell type (*Figure 3D*). Together, this work illustrates that a CNN approach can be used to distinguish diverse cell types within a given image quality window.

## Nucleocentric predictions remain accurate regardless of culture density

Morphological profiling relies on accurate cell detection. This may become difficult in dense cultures such as iPSC-cultures, clustered cells, tissues, and tissue-mimics. Having established a method to distinguish cell types with high accuracy, we next asked how robust the classification was to increasing cell density. To this end, we acquired images of 1321N1 and SH-SY5Y monocultures, grown at densities ranging from 0 to 100% confluency (*Figure 4A*). Based on the nuclear count, we binned individual fields into 6 density classes (0–20%, 20–40%, 40–60%, 60–80%, 80–95%, 95–100%) and trained a CNN with equal sampling of cell numbers per density class to avoid bias. No decrease in accuracy was observed until the culture density reached 80% confluency. At very high densities (95–100%), we found a significant decrease in the prediction accuracy (F-score: 0.92±0.05) (*Figure 4B*). We reasoned that under these conditions cell shape would be predominantly determined by neighbouring cells and cell segmentation performance would decrease (*Figure 4—figure supplement 1*). Nuclei are less malleable and even in dense cultures remain largely separated, allowing their robust segmentation and avoiding CNN misclassifications resulting from segmentation errors. Hence, we asked whether using the nuclear ROI as input would improve classification performance at high densities. However, despite the relatively high average F-score of 0.91±0.05 (*Figure 4C*), the performance was consistently lower than whole cell ROI across the density range and the performance still decreased with full confluency. To understand these results, we inspected the GradCAM output for these predictions and found that an important part of the attention of the CNN was diverted to the background (*Figure 4D*, *Figure 4—figure supplements 2–7*). We interpret this result as the CNN using the background as a proxy for nuclear size. To rule out bias by setting the background to zero in nuclear crops, the CNN was also trained on the same crops with randomly speckled backgrounds, but despite a shift in attention to the nucleus, similar prediction performance was attained (*Figure 3—figure supplement 1*). The classification performance was not biased by segmentation errors (*Figure 4—figure supplement 8A*) but may be influenced by the fact that the nuclear area is affected by culture density (*Figure 4—figure supplement 8B*). In highly dense cultures, the dynamic range of the nuclear size reduces as all nuclei become more compact and the nuclear size range decreases (*Figure 4—figure supplement 8B*). Thence, we tested an intermediate condition which exploits the more robust nuclear segmentation but also includes part of the (sub-)cellular local surrounding information as input (*Figure 4—figure supplement 8C*). To identify the optimal patch size, we varied the size of a square box centred around the nuclear centroid from 0.6 to 150 μm (*Figure 4E*). Within a range of 12–18 μm, we found

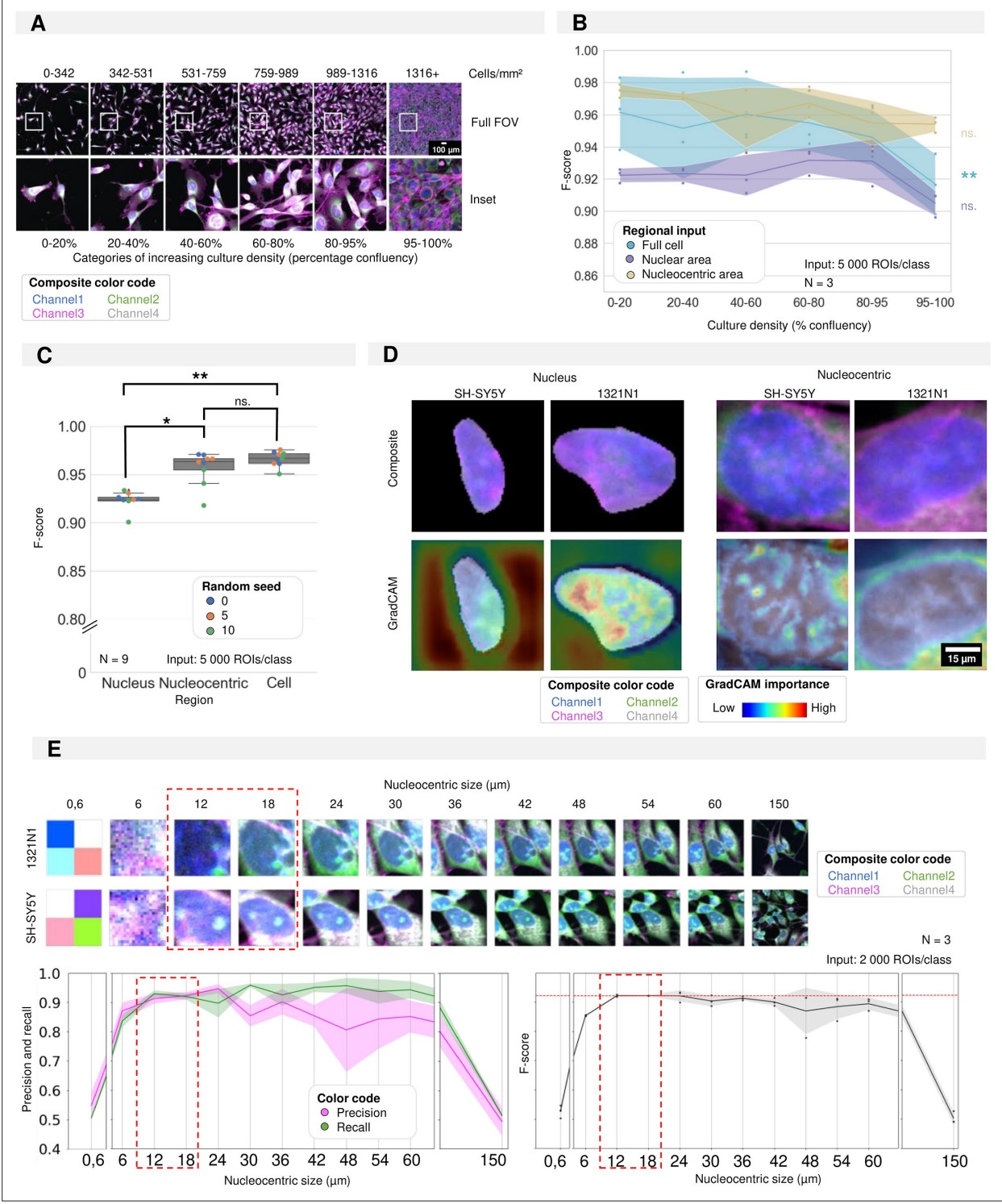

**Figure 4.** Required regional input for accurate convolutional neuronal network (CNN) training in cultures of high density. (**A**) Selected images and insets for increasing culture density with the density categories used for CNN training. (**B**) Results of three CNN models trained using different regional inputs (the full cell, only nucleus, or nucleocentric area) and evaluated on data subsets with increasing density. Each dot represents the F-score of one classifier (model initialisation, N=3) tested on a density subset. Classifiers were trained with the same random seed. Ribbon represents the standard deviation. (**C**) CNN performance (F-score) of CNN models using different regional inputs (full cell, only nucleus, or nucleocentric area). Each boxplot represents three model initialisations for three different random seeds (N=9). (**D**) Images of example inputs for both the nuclear and nucleocentric regions. The composite image contains an overlay of all cell painting (CP) channels (top). The GradCAM image overlays the GradCAM heatmap on top of the composite image, highlighting the most important regions according to the CNN (bottom). One example is given per cell type. (**E**) Systematic in- and

*Figure 4 continued on next page*

*Figure 4 continued*

decrease (default of 18 μm used in previous panels) of the patch size surrounding the nuclear centroid used to determine the nucleocentric area. Each dot represents the results of one classifier (model initialisation, N=3). Ribbon represents the standard deviation. The analysis was performed using a mixed culture dataset of 1321N1 and SH-SY5Y cells (*Figure 5*).

The online version of this article includes the following figure supplement(s) for figure 4:

**Figure supplement 1.** Segmentation performance as a function of cell density.

**Figure supplement 2.** GradCAM maps per region and density for cultures of less than 20% confluency.

**Figure supplement 3.** GradCAM maps per region and density for cultures between 20–40% confluency.

**Figure supplement 4.** GradCAM maps per region and density for cultures between 40–60% confluency.

**Figure supplement 5.** GradCAM maps per region and density for cultures between 60–80% confluency.

**Figure supplement 6.** GradCAM maps per region and density for cultures between 80–95% confluency.

**Figure supplement 7.** GradCAM maps per region and density for cultures higher than 95% confluency.

**Figure supplement 8.** Required regional input for CNN training in cultures of increasing density.

a maximal F-score of 0.96±0.02 (*Figure 4E*). Further increasing the nucleocentric patch size did not majorly affect the F-score but significantly decreased the precision and variability of recall for CNN predictions (*Figure 4E*). Hence, for further experiments, a nucleocentric patch size of 18 μm was used. GradCAM images revealed that this latter approach led the CNN to focus on perinuclear structures (*Figure 4D* and *Figure 4—figure supplements 2–7*). Interestingly, when using this nucleocentric approach, the prediction performance was maintained at almost confluent cell densities in contrast with whole cell approaches (*Figure 4B*). Thus, we conclude that using a nucleocentric region as input for the CNN is a valuable strategy for accurate cell type identification in dense cultures and that this method is not very sensitive to the patch size.

## Cell prediction remains accurate in mixed cell culture conditions

Although a very high classification accuracy was obtained with nucleocentric CNN predictions, both training and testing were performed with input images drawn from monocultures. As our goal was to allow cell type prediction in complex, heterogeneous cultures, we next switched to a more faithful situation in which we co-cultured both cell types. Ground-truth for these predictions was generated by either performing post-hoc immunofluorescence (IF) staining with cell-type specific antibodies (CD44 for 1321N1 and TUBB3 for SH-SY5Y cells), or by differential replication labelling (i.e. by incubating the two cell types with EdU and BrdU, respectively, prior to mixing them), after dye quenching (*Radtke et al., 2022*; *Figure 5A*). Replication labelling proved significantly more successful for binary ground truth classification (through intensity thresholding) than IF staining for the cell lines we used (*Figure 5B*). When training a CNN to recognize the cell types using these markers as ground truth, we found that the prediction accuracy on the left-out dataset of the co-culture (Co2Co) was almost as high as when a model was trained and tested on monocultures (Mono2Mono) (*Figure 5C*). We then tested whether it was possible to train a classifier based on monocultures only, for cell type prediction of cells in co-culture (Mono2Co). This resulted in an F-score of 0.86±0.01%. This drop in performance may be caused by the effect on cell phenotype exerted by the presence of other cell types in mixed cultures which is not captured in monocultures. As the images from monocultures and co-cultures were obtained from different plates, we suspected inter-replicate variability in culture and staining procedures to contribute in part to the lesser performance. Therefore, we tested whether we could improve the performance of the CNN by including monocultures from the same plate as the co-cultures. This finetuning indeed improved the average performance to 0.88±0.01%, and more importantly, it significantly reduced the variability (coefficient of variation) of the predictions making it more reproducible (*Figure 5C*). Thus, while not yet reaching the same accuracy, it proves that it is possible to establish a model that recognises cell types in co-cultures while only having seen monoculture training data.

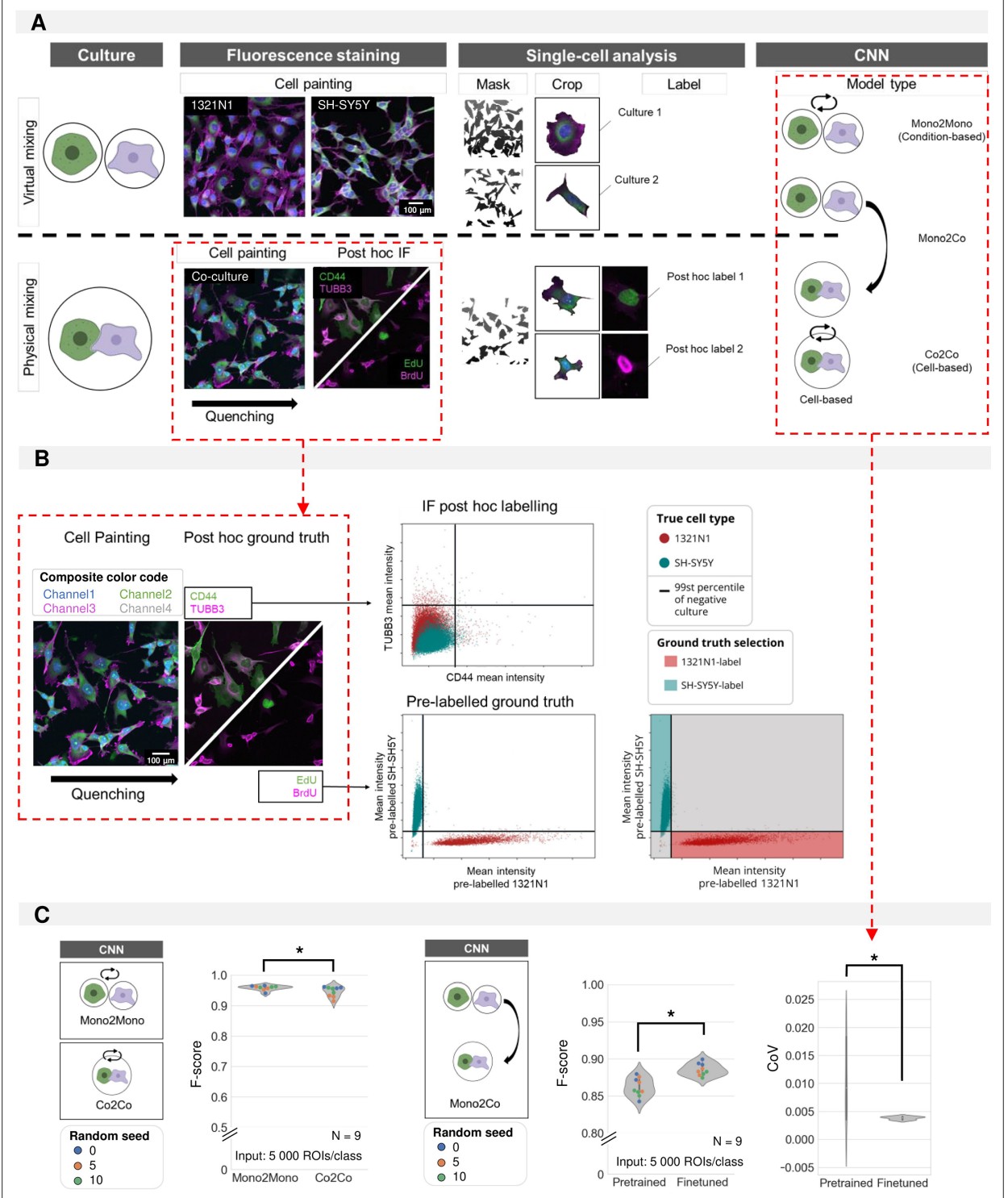

**Figure 5.** Cell type prediction in mixed cultures by post-hoc ground truth identification. (**A**) Schematic overview of virtual vs. physically mixed cultures and the subsequent convolutional neuronal network (CNN) models. For virtual mixing, individual cell types arise from distinct cultures, the cell type label was assigned based on the culture condition. For physical mixing, the true phenotype of each crop was determined by intensity thresholding after post-hoc staining. Three model types are defined: Mono2Mono (culture-based label), Co2Co (cell-based label), and Mono2Co (trained with culture-based label and tested on cell-based label). (**B**) Determining ground-truth phenotype by intensity thresholding on immunofluorescence staining (IF) (top) or pre-label (bottom). The threshold was determined on the intensity of monocultures. Only the pre-labelled ground-truth resulted in good cell type separation by thresholding for 1321N1 vs. SH-SY5Y cells. (**C**) Mono2Mono (culture-based ground truth) vs. Co2Co (cell-based ground truth) models for cell type classification. Analysis performed with full cell segmentation. Mann-Whitney-U-test p-value 0.0027. Monoculture-trained models were tested on

*Figure 5 continued on next page*

*Figure 5 continued*

mixed cultures. Pretrained models were trained on independent biological replicates. These could be finetuned by additional training on monoculture images from the same replicate as the coculture. This was shown to reduce the variation between model iterations (Median performance: Mann-Whitney-U-test, p-value 0.0015; Coefficient of variation: Mann-Whitney-U-test, p-value 3.48e-4). Each dot in the violinplots represents the F-score of one classifier (model initialisation, N=9). Classifiers were trained 3 x with three different random seeds.

The online version of this article includes the following figure supplement(s) for figure 5:

**Figure supplement 1.** Fluorescence quenching over time using LiBH4.

**Figure supplement 2.** Methods.

## Cell-type profiling can be applied to stage iPSC-derived neuronal cultures

iPSC-derived neuronal cell cultures suffer from significant inter- and intra-batch variability and could benefit from an efficient quality control (*Hernández et al., 2022*; *Volpato and Webber, 2020*). Thence, we applied our nucleocentric phenotyping pipeline to stage the maturity of a neuronal cell culture based on its cell type composition. Using a guided protocol (*Bell et al., 2019*), two differentiation stages were simulated: a primed stage, where most cells are assumed to be cycling neural progenitors (NPCs), and a differentiated stage where most cells are post-mitotic neurons (*Figure 6A*). The two cell types were discriminated by post-hoc IF staining for the cell cycle marker Ki67 (NPC) and the microtubule marker ß-III-tubulin (TUBB3, neurons) (*Figure 6B*). Not all cells in the CP image could be assigned with a ground truth label due to cell detachment upon re-staining or sheer absence of the tested marker. Since no single monoculture consists of 100% of either cell type, we applied gates to retain only those cells that showed unequivocal staining for either one of both markers. Based on these gates, ROIs were either classified as neuron (Ki67-/TUBB3+), NPC (Ki67+/TUBB3-), or undefined (outside of gating boundaries). We assume the latter category to represent transitioning cells in intermediate stages of neural development, un- or dedifferentiated iPSCs. This gating strategy resulted in a fractional abundance of neurons vs. total (neurons+NPC) of 36.4% in the primed condition and 80.0% in the differentiated condition (*Figure 6C*). In a first attempt to classify cells within the guided culture, we trained a CNN on individual cell inputs using the culture condition (primed or differentiated) as ground truth, which given the heterogeneity, can be considered as a weak label. This resulted in an F-score of 0.86±.,01%. When we used the cell-level IF ground truth labels instead, we obtained a classification performance of 0.96±0.00%. For comparison, a shallow learner (RF), showed a significantly lower F-score of 0.87±0.02 (*Figure 6D*). Applying this cell-based (as opposed to condition-based) CNN to the two culture conditions resulted in a predicted fractional abundance of neurons to NPC of 40,5% in images of the primed condition and 74.2% in images of the differentiated condition – aligning well with the manually defined ratios (*Figure 6E*). Both supervised and unsupervised UMAP dimensionality reduction on the feature embeddings of the cell-based classifier revealed a clear clustering of both phenotypes suggesting that the CNN captures the differences in morphotextural fingerprint between neurons and NPCs well (*Figure 6F*). We then went on to evaluate the established cell-based classifier to a primed neuronal cell culture undergoing gradual spontaneous differentiation after dual SMAD inhibition *Shi et al., 2012*. We examined cell cultures at 13, 30, 60, and 90 days in vitro (DIV) after the start of the differentiation process and visually confirmed a gradual change in neural maturity by the progressive increase of cells with smaller somas and long, thin ramifications (*Figure 6G*). The slower shift in the neuron-to-NPC fractional abundance with increasing time in this spontaneous differentiation setting was confirmed by the cell-based CNN model trained on the guided differentiation dataset *Figure 6H*. This illustrates the generalizability and transferability of cell type profiling for iPSC-derived neuronal culture staging.

## Cell type identification can be applied to mixed iPSC-derived neuronal cultures regardless of the activation state

Next to NPCs and neurons, iPSC-derived neuronal cultures are often studied in conjunction with other relevant cell types that influence neuronal connectivity and homeostasis such as astrocytes and microglia (*Kuijlaars et al., 2016*). Therefore, we tested whether the cell-based approach could be extended to these cell types as well. We generated monocultures of iPSC-derived astrocytes, neurons, and microglia from the same parental iPSC line (*Figure 7A*) and trained a nucleocentric CNN to

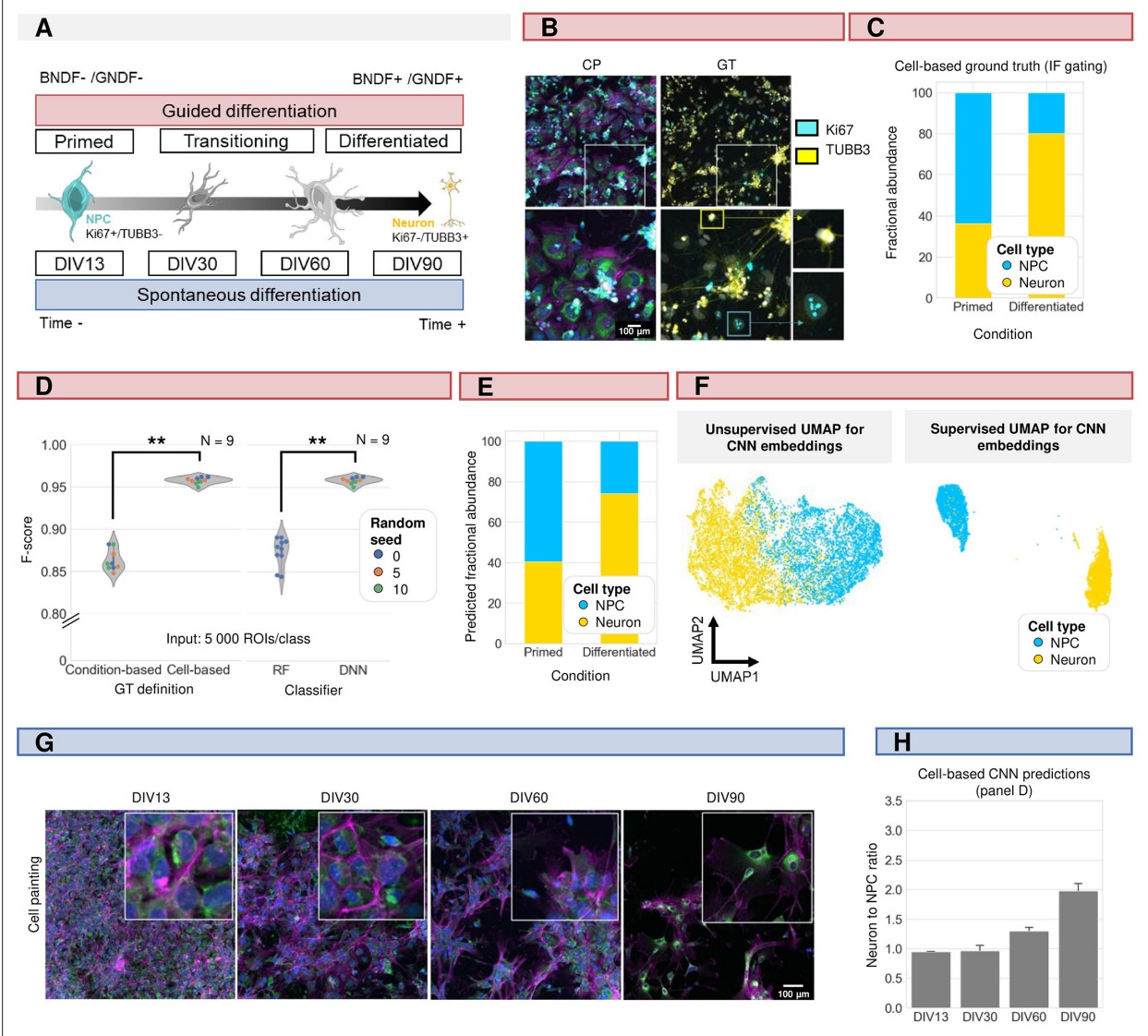

**Figure 6.** Induced pluripotent stem cell (iPSC)-derived differentiation staging using morphology profiling. (**A**) Schematic overview of guided vs. spontaneous neural differentiation. DIV = days in vitro. Selected timepoints for analysis of the spontaneously differentiation culture were 13, 30, 60, and 90 d from the start of differentiation of iPSCs. (**B**) Representative images of morphological staining (colour code as defined in *Figure 1A*) and post-hoc immunofluorescence staining (IF) of primed and differentiated iPSC-derived cultures (guided differentiation). Ground-truth determination is performed using TUBB3 (for mature neurons) and Ki67 (mitotic progenitors). (**C**) Fraction of neurons vs. neural progenitor (NPC) cells in the primed vs. differentiated condition as determined by IF staining. Upon guided differentiation, the fraction of neurons increased. (**D**) Left: CNN performance when classifying neurons (Ki67-/TUBB3+) vs. NPC (Ki67+/TUBB3-) cells using either a condition-based or cell-type-based ground truth. Each dot in the violinplots represents the F-score of one classifier (model initialisation). Classifiers were trained with different random seeds. Mann-Whitney-U-test, p-value 4.04e-4. Right: comparison of CNN vs. RF performance. Mann-Whitney-U-test, p-value 2.78e-4. (**E**) Fractional abundance of predicted cell phenotypes (NPC vs. neurons) in primed vs. differentiated culture conditions using the cell-based CNN. (**F**) Unsupervised and supervised UMAP of the cell-based CNN feature embeddings. Plot colour-coded by cell type. Points represent individual cells. (**G**) Representative images of spontaneously differentiating neural cultures. Colour code as defined in *Figure 1A*. (**H**) Prediction of differentiation status using the cell-based CNN model trained on guided differentiated culture.

identify each cell type. This led to a prediction accuracy of 96.81±0.95% (*Figure 7A*). Recognizing that these cell types are visibly morphologically distinct, we tested the robustness of the model to experimental perturbations in cell state (*Figure 7B*). To this end, we induced reactivity in the iPSC-derived microglial culture by LPS treatment. A CNN trained on monocultures of neurons, unchallenged or LPS-treated microglia showed high accuracy in differentiating neurons from microglia (98% accurate), although the difference between reactive and resting-state microglia in this tripartite model proved

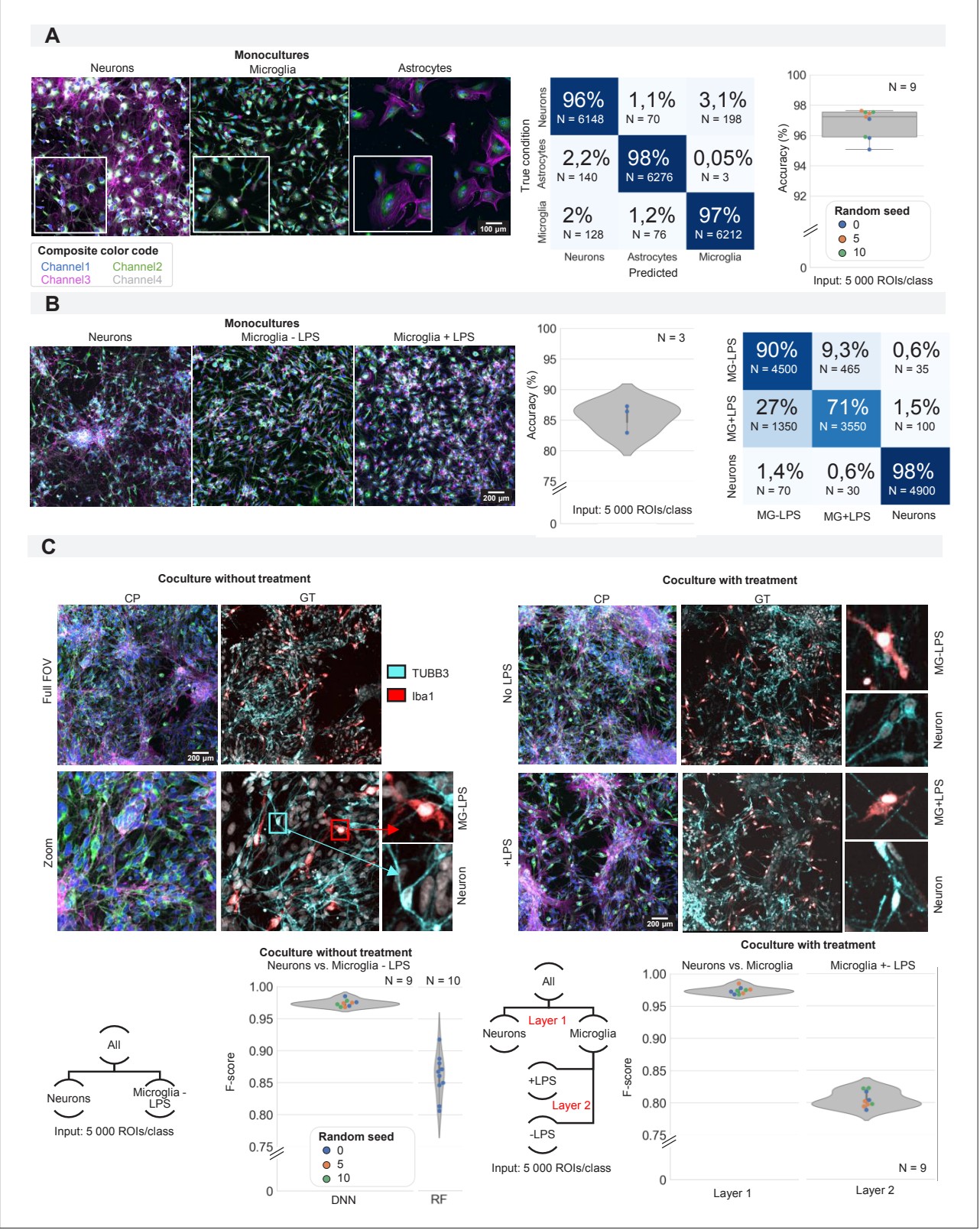

**Figure 7.** Induced pluripotent stem cell (iPSC) cell type identification using morphology profiling. (**A**) Representative images of iPSC-derived neurons, astrocytes, and microglia in monoculture with morphological staining. Colour code as defined in *Figure 1A*. Prediction accuracy of a convolutional neuronal network (CNN) trained to classify monocultures of iPSC-derived astrocytes, microglia, and neurons with confusion matrix (average of all models). Each dot in the boxplot represents the F-score of one classifier (model initialisation, N=9). Classifiers were trained 3 x with three different

*Figure 7 continued on next page*

Figure 7 continued

random seeds. (**B**) Representative images of monocultures of iPSC-derived neurons and microglia treated with LPS or control. Colour code as defined in **Figure 1A**. Prediction accuracy and confusion matrix (average of all models) are given. Each dot in the violinplot represents the F-score of one classifier (model initialisation, N=3). (**C**) Representative images of a mixed culture of iPSC-derived microglia and neurons. Ground-truth identification was performed using immunofluorescence staining (IF). Each dot in the violinplot represents the F-score of one classifier (model initialisation). Classifiers were trained three different random seeds. Results of the CNN are compared to shallow learning (random forest, RF). The same analysis was performed for mixed cultures of neurons and microglia with LPS treatment or control. A layered approach was used where first the neurons were separated from the microglia before classifying treated vs. non-treated microglia. Each dot in the violinplot represents the F-score of one classifier (model initialisation, N=9). Classifiers were trained 3 x with three different random seeds.

more challenging (71% and 90% accuracy, resp.). Repeating those experiments in mixed cultures of neurons and microglia (using Tubb3, resp. Iba1 as ground-truth IF markers), yielded an F-score of 0.98±0.01 for the classification of neurons vs. microglia (for comparison, a classical RF approach only returned an F-score of 0.86±0.03) (**Figure 7C**). In the presence of LPS, the CNN model yielded a high F-score of 0.97±0.01 (**Figure 7C**). This implies that the shift in cell state does not significantly affect the CNN's ability to distinguish neurons from microglia. When implementing a tiered approach, in which after neuron-microglia recognition a second model was tasked with the classification of reactive vs. resting state microglia without the presence of neurons, this phenotype proved more challenging for the CNN to predict, as evidenced by an F-score of 0.80±0.01 (**Figure 7C**). Based on these results, we conclude that nucleocentric phenotyping can be used to gauge the cell type composition of iPSC-derived cultures. Different states of individual cell types can also be recognized albeit with lower performance.

## Discussion

iPSC-derived cell cultures have the potential to improve the translatability of biomedical research, but represent a challenge due to their variability and multicellular composition. With our work, we have developed a method to identify neural cell types in mixed cultures to aid in their validation and application in routine screening settings. We first benchmarked our approach using neural cell lines and found that a CNN-based approach outperforms shallow learning (RF) based on handcrafted feature extraction, even with a limited number of input images. This aligns well with recent data showing CNN-based feature representations outperform shallow techniques for condition-based drug response prediction (**Dincer et al., 2018**). To assess the robustness and sensitivity of deep learning for cell-type prediction, we evaluated the CNN performance as a function of input data size and quality. Next, we tested the performance of predictions for dense single and multi-cellular cultures. Finally, we assessed its performance in the presence of different cell states.

Although benchmarking revealed a minimal image resolution of 0.6 µm/pixel and SNR of approximately 10 dB for optimal CNN performance, lower-quality images may still yield acceptable performance, especially with the use of advanced image restoration algorithms (**Weigert et al., 2018**). We found that all channels contribute to the overall prediction accuracy but that the whole cell is not required to obtain the highest performance. In line with earlier studies that highlight the biomarker potential of the nucleus for patient stratification (**De Vos et al., 2010**), cell state identification (**Heckenbach et al., 2022**), or drug response (**Cimini et al., 2023**), we found that the nuclear ROI as such already carries highly distinctive information for cell type prediction. However, extension to its direct surroundings proved to be the best and most robust input across a range of cell densities. The nucleocentric approach is based on more robust nuclear segmentation, reducing segmentation errors whilst still retaining input information from the structures directly surrounding the nucleus. At higher cell density, the whole-cell segmentation becomes more error-prone, while also losing morphological information (**Figure 4—figure supplement 1**). The nucleocentric approach is more consistent as it relies on a more robust segmentation and does not blank the surrounding region. This way it also buffers for occasional nuclear segmentation errors (e.g. where blebs or parts of the nucleus are left undetected). GradCAM maps on nucleocentric crops highlight specifically those structures surrounding the nucleus (reflecting ER, mitochondria, Golgi) indicating their importance in correct cell classification. Stepwise in- and decreasing the size of the nucleocentric window in a range between 12 and 60 µm revealed that this approach is not sensitive to alterations in patch size. We therefore conclude that the superior

performance of the nucleocentric approach in dense cultures stems from a reduced susceptibility to segmentation errors. It opens possibilities for applying cell profiling in the future to even more complex 3D cell systems such as tissue or organoid slices, where accurate cell segmentation becomes extremely challenging. Indeed, our results indicate that it might be possible to discriminate individual cells in extremely crowded environments. For specific applications such as the classification of healthy vs. tumour tissue, binary classification models can prove useful. It has been shown that a dual-task graph neural network can classify epithelioid and sarcomatoid tumor cells with the cellular resolution for accurate mesothelioma subtype determination (*Eastwood et al., 2023*). However, the diversity of cell types and intermediates present in tissue may complicate classification and demand richer ground truth to refine such predictions. In the future, it may even be possible to include volumetric information, but this will require optimisation of the sample preparation procedures as well. To increase the accuracy, one could resort to intelligent data augmentation (*Mikolajczyk and Grochowski, 2018*) or transfer learning (*Nguyen et al., 2018*) strategies.

One crucial realisation of this work is that cell types can be identified in mixed cultures solely using the input from monocultures. This implies that cells retain their salient characteristics even in more complex heterogeneous environments. While for now, predictions are still superior when training directly on mixed cultures, we found that the prediction accuracy of monoculture-trained models can be increased by employing replicate controls. This suggests that it may become possible to apply or adapt existing models without the need for a cell-level ground truth as provided by post-hoc labelling. This technique could potentially be of use for cultures or tissues where no antibody- or pre-labelling is possible (e.g. no unique IF marker is available, non-replicating cells). While a ground-truth independent method holds promise for bulk predictions (e.g. for quality control purposes), the use of post-hoc labelling allows the building of more refined models that can distinguish multiple cell types, and/or cell states at once. Especially using cyclic staining or spatial transcriptomics, much richer information content can be gained. Multiplexed imaging has previously shown its utility in gaining in-depth information on culture composition and differentiation status in iPSC-derived neural cultures progenitor, radial glia, astrocytes (immature and mature) and neurons (inhibitory and excitatory) (*Tomov et al., 2021*). Similarly, this could be expanded to cell states (apoptosis, DNA damage), although current prediction performance strongly varies with the phenotype (*Way et al., 2021*).

Applying the method to iPSC-neuronal cultures revealed its potential to score their differentiation state. Although guided protocols managed to speed up the differentiation process and lead to enhanced culture purity, the neural differentiation process was not binary, as evidenced by the mixed presence of Ki67+/TUBB3- and Ki67-/TUBB3 + cells. The spontaneous differentiation protocol on the other hand illustrated the unpredictable nature and slow nature of the differentiation process. Many groups highlight the difficulty of reproducible neural differentiation and attribute this to culture conditions, cultivation time, and variation in developmental signalling pathways in the source iPSC material (*Strano et al., 2020*; *Galiakberova et al., 2023*). Spontaneous neural differentiation typically requires around 80 days before mature neurons arise that show circuit activity (*Shi et al., 2012*), but the exact timing and duration vary. This variation negatively affects the statistical power when testing drug interventions and thus prohibits the application of iPSC-culture derivatives in routine drug screening. We have shown that CP captures the morphological changes associated with neural differentiation, allowing us to reliably predict the gradual transition from NPC to a more differentiated state. We recognize that differentiating iPSC cultures are highly heterogenous and are composed of a landscape of transitioning progenitors in various stages of maturity that our binary models currently fail to capture. We filtered out these cells with a 'dubious' IF profile as they would negatively affect the model by introducing noise. In future iterations, one could envision defining more refined cell (sub-) types in a population based on richer post-hoc information (e.g. through cyclic immunofluorescence or spatial single-cell transcriptomics). While we emphasize the value of identifying fixed states as a fast-track to gather the composition of an iPSC-derived culture, an equally valuable angle consists of embracing the continuum of states that such cultures represent. Here, predictions would then demand regression-based approaches (*Du and Xu, 2017*). Pioneering efforts using live-cell imaging and machine learning have allowed predicting gradual cell state transitions, for example in the context of myoblast, adipocyte, or osteoblast differentiation (*Shakarchy et al., 2024*; *Mai et al., 2023*). Label-free timelapse imaging has also shown potential in this context (*Shakarchy et al., 2024*). Given that many of the cell painting dyes are compatible with live cells, it is conceivable that our approach is

amenable to further refining the assessment of iPSC differentiation status in real-time, e.g., with the inclusion of brightfield or phase contrast images.

After testing the CNN performance on heterogenous cultures, we have added an additional layer of complexity by inducing microglial reactivity in a coculture of neurons and microglia. We found that we could still predict the cell type regardless of the treatment. The increased variability within the microglial subpopulation did not impact the CNNs' ability to discriminate cell types in mixed culture. Furthermore, using the layered approach, the resting-state and reactive subgroups within the microglial population could also be classified, albeit with a lesser prediction accuracy. This could be explained by the fact that microglia grown in vitro are not completely homeostatic, even in the absence of LPS (*Gosselin et al., 2017*), and may require different cell types to adopt a more natural state. Having more faithful mixed cultures with cells in their endogenous states, and an apt tool to recognize both cell type and state, holds promise to enhance the relevance of preclinical screens, to increase the accuracy of drug targets, and ultimately, to lead to more precise therapeutic strategies (*Engle et al., 2018*). For now, we have tailored models to the individual datasets, but it is conceivable that a more generalized CNN could be established for multiple culture types. This would obviously demand a much larger dataset to encompass the variability encountered in such models (e.g. various starting iPSC lines, and various differentiation protocols). Publicly available datasets (e.g. JUMP-cell painting consortium) can aid in creating an input database containing a large variability (different iPSC lines, different neural differentiation protocols, …), which would ultimately lead to a more robust predictor. Our results showing the prediction accuracy of a guided differentiation model on spontaneously differentiating cultures indicate that the approach can be transferred to other differentiation protocols as well. Inclusion of more input images and variability should thus enable the developing of a generalist model for other differentiation protocols without the need for ground truth validation and further CNN training.

In conclusion, we have developed a novel application for unbiased morphological profiling by extending its use to complex mixed neural cultures using sequential multispectral imaging and convolutional network-informed cell phenotype classification. We show that the resulting predictors are robust with respect to biological variation and cell culture density. This method holds promise for use in the quality control of iPSC cultures to allow their routine use in high-throughput and high-content screening applications.

## Methods
### Cell culture

Cells were cultured at 37 °C and 5% $CO_2$. 1321N1 (RRID:CVCL_0110; cat. nr.: 86030402) and SH-SY5Y (RRID:CVCL_0019; cat.nr.: 94030304) cell lines were maintained in DMEM-F12 +Glutamax (Gibco, 10565018) supplemented with 10% Fetal Bovine Serum (Gibco, 10500064). Their identity has been authenticated by the supplier (ATCC) and regular mycoplasma checks were performed. Cell seeding prior to imaging was done in 96-well black multiwell plates with #1.5 glass-like polymer coverslip bottom (Cellvis, P96-1.5P). Only the inner 60 wells were used, while the outer wells were filled with PBS-/- to avoid plate effects. Plates were coated with Matrigel (Corning, 734–1440) After seeding, the imaging plate was centrifuged at 100 g for 3 min.

iPSCs (Sigma-Aldrich, iPSC0028 Epithelial-1; RRID:CVCL_EE38) were cultured on Matrigel (Corning, 734–1440) in Essential 8 medium (Gibco, A1517001). Their identity has been authenticated by the supplier (Sigma-Aldrich) and regular mycoplasma checks were performed. Upon cell thawing, Rock inhibitor (Y-27632 dichloride, MedChem, HY-10583) was added in 10 µM concentration. Subculturing of iPSCs was performed with ReLeSR (Stemcell Technologies, 05872).

Unguided differentiation *Shi et al., 2012* of iPS cells to neural progenitor cells (NPCs) was started by subculturing the iPSCs single cell using Tryple Express Enzyme (Life technologies, 12605010) at a density of 10e4 cells/cm² in mTesR1 medium (Stemcell Technologies, 85850) and Rock inhibitor. The following day, differentiation to NPCs was started by dual-SMAD inhibition in neural maintenance medium (1:1 Neurobasal (Life technologies, 21103049):DMEM-F12 + Glutamax (Gibco, 10565018), 0.5 x Glutamax (Gibco, 35-050-061), 0.5% Mem Non Essential Amino Acids Solution (Gibco, 11140050), 0.5% Sodium Pyruvate (Gibco, 11360070), 50 µM 2-Mercaptoethanol (Gibco, 31350010), 0.025% Human Insulin Solution (Sigma-Aldrich, I9278), 0.5X N2 (Gibco, 17502048), B27 (Gibco, 17504044),

50 U/ml Penicillin-Streptomycin (Gibco, 15140122)) supplemented with 1 µM LDN-193189 (Miltenyi, 130-106-540), SB431542 (Tocris, 1614). This dual blockade of SMAD signalling in iPSCs is induces neural differentiation by synergistically causing the loss of pluripotency and push towards neuroecto-dermal lineage (*Chambers et al., 2009*). Daily medium changes were performed for 11 consecutive days. Following neural induction, neural rosettes were selected by STEMdiff Neural Rosette Selection Reagent (Stemcell technologies, 05832). Maintenance of neural progenitor cells was performed in a neural maintenance medium with 10 µM bFGF (Peprotech, 100–18 C) added after subculturing. Single-cell detachment of NPCs during maintenance was performed using Tryple Express Enzyme. Cell seeding prior to imaging is done in 96-well black multiwell plates µCLEAR (Greiner, 655090) coated with Poly-L-ornithine (Sigma-Aldrich, P4957) and laminin (Sigma-Aldrich, L2020). Only the inner 60 wells were used, while the outer wells were filled with PBS-/- to avoid plate effects. After seeding, the imaging plate was centrifuged at 100 g for 3 min.

Guided iPSC differentiation to neurons was performed according to *Bell et al., 2019*. The initial phase of neural induction consisted of 12 d neural induction medium (DMEM-F12 + Glutamax (Gibco, 10565018), 1x N2 (Gibco, 17502048), 1 x B27 (Gibco, 17504044),1mg/ml BSA (Sigma-Aldrich, A7979), 0.5% Mem Non-Essential Amino Acids Solution (Gibco, 11140050)). Of these 12 d, the first 7 were supplemented with cytokines for dual-SMAD inhibition (1 µM LDN-193189 (Miltenyi, 130-106-540), SB431542 (Tocris, 1614)). Following neural induction, the NPCs were floated in uncoated MW6 culture plates in NPC medium (DMEM-F12 + Glutamax (Gibco, 10565018), 1x N2 (Gibco, 17502048), 1 x B27 (Gibco, 17504044), 10 µM bFGF (Peprotech, 100–18 C), 10 µM EGF (Peprotech, 100–47)). After 4 d, NPC clusters of appropriate size were filtered using a cell strainer (37 µm cell strainer, Stemcell technologies, 27250) and plated on Matrigel-coated (Corning, 734–1440) MW6 culture plates. NPCs can now be expanded in the NPC medium. Guided differentiation into forebrain neurons can be induced by switching to neuronal medium BrainPhys (STEMCELL Technologies, 05792), 1x N2 (Gibco, 17502048), 1 x B27 (Gibco, 17504044), 10 µM BDNF (PeproTech, AF-450–02), 10 µM GDNF (Pepro-Tech, AF-450–02) for 15 d before fixation.

Differentiation of iPSC to microglia (*Haenseler et al., 2017*) was performed by the formation of embroid bodies (EBs) with 10e3 iPSCs/well in a 96-well U-bottom plate (Corning, 351177) coated with Anti-Adherence Rinsing Solution (Stemcell technologies, 07010) in mTeSR medium supplemented with Rock inhibitor, 50 ng/mL BMP4 (Peprotech, 120–05), 50 ng/mL VEGF (Peprotech, 100–20), 20 ng/mL SCF (Peprotech, 250–03). 75% medium is changed for four consecutive days. After meso-derm induction, EBs are transferred to a six-well plate with 20 EBs/well and placed in macrophage precursor medium (X-vivo15 (Lonza, BE02-060Q), 100 ng/mL M-CSF (Peprotech, 300–25), 25 ng/mL IL-3 (Peprotech, 213–13), 1 x Glutamax, 50 U/ml Penicillin-Streptomycin, 50 µM 2-Mercaptoethanol). 14 d after macrophage differentiation, macrophage precursors were harvested using a cell strainer (Stemcell technologies, 27250). Macrophage precursors were added to the NPC culture in 1:1 neural maintenance medium:microglia medium (DMEM-F12 + Glutamax, 100 ng/mL M-CSF (Peprotech, 300–25), 100 ng/mL IL-34 (Peprotech, 200–34), 1 x Glutamax, 50 U/ml Penicillin-Streptomycin, 50 µM 2-Mercaptoethanol).

## Replication labelling

Prior to co-seeding of 1321N1 and SH-SY5Y mixed cultures, individual cultures were incubated with either 10 µM EdU (Click-iT EdU Imaging Kit, Life Technologies, C10340) or 10 µM BrdU (Sigma-Aldrich, B5002) for 24 h. This incubation time exceeded the doubling time, allowing incorporation of the nucleotide analog in all cells. This labelling period was followed by a 24 hr washout period in a regular cell culture medium. After washout, the cells were subcultured and plated in coculture. In half of the replicate, SH-SY5Y cells received BrdU while 1321N1 cells received EdU. For the remainder of the wells, the pre-label switched cell types.

## Morphological staining

Morphological staining (cell painting) was adapted from *Bray et al., 2016*. After careful titration, all dye concentrations were adjusted and evaluated for compatibility with the 4-channel laser and filter combinations available on the confocal microscope (see further). Staining was performed on cell cultures fixed in 4% PFA (roti-histofix 4% paraformaldehyde, Roth, 3105.2) for 15 min. Cells were rinsed once with PBS-/- (Life Technologies, 10010015) prior to fixation and 3×5 min post-fixation. Staining

**Table 2.** Specifications of morphological staining composition.

**Staining solution 1**

| Dye | Target | Excitation (nm) | Emission (nm) | Stock concentration | End dilution | Supplier | Catalog # |
|---|---|---|---|---|---|---|---|
| DAPI | Nucleus | 350 | 470 | 2.5 mg/ml | 1/500 | Sigma Aldrich | D9542 |
| Concanavalin | ER | 480 | 490–540 | 5 mg/ml | 1/300 | Life Technologies | C11252 |
| WGA | Golgi | 550 | 560–570 | 1 mg/ml | 1/600 | Life Technologies | W32464 |
| Phalloidin | Actin | 570 | 580–610 | 66 µM | 1/1000 | Life Technologies | A12380 |
| Mitotracker | Mitochondria | 640 | 650–660 | 1 mM | 1/1000 | Life Technologies | M22426 |
| **Staining solution 2** | | | | | | | |
| Syto14 | Nucleoli/RNA | 525 | 540–580 | 5 mM | 1/1000 | Life Technologies | S7576 |

solutions were prepared fresh before staining in PBS-/- with 0.3% Triton-X-100 (Sigma-Aldrich, X100) (*Table 2*). Each staining solution was incubated for 30 min on a shaker at RT in the dark. After staining, the cells were washed 1 x with PBS -/- and sealed for imaging.

## Cyclic staining and immunocytochemistry

Cyclic staining is executed by fluorescence quenching after each sequential imaging round. 1 mg/ml in ddH$_2$O LiBH$_4$ solution (*Radtke et al., 2022*) (Acros Organics, 206810050) was prepared fresh before use. 1.5 hr incubation of quenching solution was performed before each successive staining series. After incubation, the quenching solution was removed by washing 3×5 min in PBS-/-. Successful fluorescence quenching was microscopically verified before proceeding with immunofluorescence staining (IF) (*Table 3*, *Figure 5—figure supplement 1*).

Cells are treated with PAV blocking buffer (Thimerosal 0.5% (Fluka, 71230), NaN3 0.1% (Merck, k 6688), Bovine serum albumin (Sigma-Aldrich, A7284), Normal horse serum, PBS-/-) for 8 min. The desired primary antibodies (pAB) are diluted in PAV blocking buffer. pAB incubation was performed 12 hr (overnight) at 4 °C after which the cells were washed 1×5 min in PBS-/- followed by incubation in secondary antibody solution (sAB) in PAV + DAPI for nuclear counterstain. sAB staining was

**Table 3.** Used antibodies.

| Antibody | | Host | Supplier | Catalog # | RRID |
|---|---|---|---|---|---|
| | Anti-BrdU | Sheep | Abcam | ab1893 | AB_302659 |
| | Anti-MAP2 | Chicken | Synaptic systems | 188006 | AB_2619881 |
| | Anti-TUBB3 | Mouse | BioLegend | 801202 | AB_2313773 |
| | Anti-CD44 | Mouse | Millipore | MAB4065 | AB_95019 |
| | Anti-GFAP | Goat | Abcam | ab53554 | AB_880202 |
| | Anti-Iba1 | Rabbit | Wako | 019–19741 | AB_839504 |
| Primary antibodies | Anti-Ki67 | Rabbit | Abcam | ab15580 | AB_443209 |
| | Anti-mouse-Cy3 | Donkey | Jackson ImmunoResearch | 715-165-150 | AB_2340813 |
| | Anti-mouse-Cy5 | Donkey | Jackson ImmunoResearch | 715-175-150 | AB_2340819 |
| | Anti-sheep-Cy3 | Donkey | Jackson ImmunoResearch | 713-165-147 | AB_2315778 |
| | Anti-sheep-Cy5 | Donkey | Jackson ImmunoResearch | 713-175-147 | AB_2340730 |
| | Anti-chicken-Cy5 | Donkey | Jackson ImmunoResearch | 703-175-155 | AB_2340365 |
| | Anti-goat-Cy3 | Donkey | Jackson ImmunoResearch | 705-165-147 | AB_2307351 |
| | Anti-Guinea pig-Cy3 | Donkey | Jackson ImmunoResearch | 706-165-148 | AB_2340460 |
| Secondary antibodies | Anti-Rabbit-Cy5 | Donkey | Jackson ImmunoResearch | 711-175-152 | AB_2340607 |

**Table 4.** Specifications of the used laser lines, excitation, and emission filters.

| Channel dimension | Laser line (nm) | Excitation filter (nm) | Emission filter (nm) |
|---|---|---|---|
| Channel 1 | 405 | | B447/60 |
| Channel 2 | 488 | | B520/35 |
| Channel 3 | 561 | | B617/73 |
| Channel 4 | 640 | Di01-T405/488/568/647−13x15 × 0.5 | B685/40 |

performed for 3 hr at RT while shaking. Prior to imaging, the cells were washed 2 x with PBS-/- and stored in PBS-/- +0,1% NaN₃.

BrdU staining was performed using IF, requiring DNA denaturation before pAB incubation. This was performed by 10 min incubation with 2 N HCl at 37 °C. HCl was neutralized with 0.1 M sodium borate buffer pH 8.5 for 30 min at RT. Cells were washed 3×5 min in PBS-/- before continuing with the general IF protocol. EdU click-it labelling was performed according to the manufacturer's instructions (Click-iT EdU Imaging Kit, Life Technologies, C10340) (*Figure 5—figure supplement 2A*).

## Image acquisition

Images were acquired using a spinning disk confocal microscope (Nikon CSU-W1 SoRa) with a 20×0.75 NA objective (Plan APO VC 20 x DIC N2) and Kinetix sCMOS camera (pinhole 50 µm; disk speed 4000 rpm; pinhole aperture 10; bit depth 12-bit, pixel size 0.325 µm²). We opted for confocal microscopy instead of widefield to overcome image quality limitations resulting from highly dense cell clusters. 96-well plates were scanned, capturing a minimum of 16 images per well spread in a regular pattern (0,8 mm spacing in x and y) across the surface of the well. If multiple z-slices were acquired (to correct for surface inclinations in the field of view), a maximum projection was performed before further analysis. Morphological images were acquired in all four channels. (*Table 4*).

**Table 5.** Python packages used for image and data analysis alongside the software version.

| Package | Version | Reference |
|---|---|---|
| pathlib | 1.0.1 | |
| numpy | 1.21.2 | *Harris et al., 2020* |
| tifffile | 2021.7.2 | |
| pytorch | 1.13.0 | *Paszke et al., 2019* |
| torchvision | 0.14.0 | |
| nd2reader | 3.3.0 | |
| matplotlib | 3.5.2 | *Hunter, 2007* |
| scikit-image | 0.19.3 | *Van der Walt et al., 2014* |
| scikit-learn | 1.1.3 | *Pedregosa et al., 2011* |
| argparse | 1.4.0 | |
| cellpose | 2.1.1 | *Stringer et al., 2021* |
| tqdm | 4.64.1 | |
| scipy | 1.9.3 | *Virtanen et al., 2020* |
| pandas | 1.5.1 | *McKinney, 2010* |
| seaborn | 0.12.1 | *Waskom et al., 2022* |
| imbalanced-learn | 0.10.0 | *Lemaître et al., 2017* |
| StarDist | 0.8.5 | *Weigert et al., 2020* |

## Software

Images were captured by the Nikon CSU-W1 confocal system in combination with the NIS elements software (RRID:SCR_014329). Image visualisation was later performed using Fiji freeware (*Rueden et al., 2017*; *Schindelin et al., 2015*). In-depth image analysis, pre-processing and machine learning for cell classification were performed using Python programming language (*Rossum and Drake, 2009*) in combination with Anaconda (*Anaconda Software Distribution, 2024*) (distribution and package managing software) and Visual Studio Code (code editor). The packages and versions used for data analysis are shown in *Table 5*.

## Image pre-processing (Figure 5—figure supplement 2B)

Note: All image and data analysis scripts are available on GitHub. See further the data availability statement.

### Cell/nuclear segmentation

Using a tailor-made script implementing the Cellpose (*Stringer et al., 2021*) (cell segmentation) or StarDist (*Weigert et al., 2020*) (nuclear segmentation) package, images were pre-processed by normalizing the intensity values of each channel between the first and 99th quantile. Individual channels or channel combinations for segmentation can be selected depending on the desired outcome mask. Resulting outputs included the segmentation mask alongside the quality control images. For Stardist implementation, hyperparameter probability was set at 0.6 and overlap at 0.3. For Cellpose segmentation, four models (cyto2) were averaged to obtain the final mask. Segmentation was performed on the composite of all CP channels. An estimation of the cell's diameter could be included to optimize cell detection. Cell segmentation was performed using Cellpose and used in all cases where the whole-cell crop was given as input to the CNN or data from the whole cell was used for feature extraction for RF (*Figures 1–4*). Nuclear segmentation was performed using Stardist and used for nuclear and nucleocentric crops (*Figures 4–7*).

### Ground truth alignment

Following sequential staining and imaging rounds, multiple images were captured representing the same cell with different markers. Lifting the plate of the microscope stage and imaging in sequential rounds after several days results in small linear translations in the exact location of each image. These linear translations need to be corrected to align morphological with ground truth image data within the same ROI. All images were aligned using Fourier-based image cross correlation on the intensity-normalized multichannel images. The alignment shift between image1 and image2 was determined using scikit-image phase cross correlation. Image2 was then moved according to the predetermined shift to align morphological with ground truth images.

### Ground truth phenotyping

The true cell phenotype was determined by the fluorescence intensity of the post-hoc immunostaining with class-specific markers or pre-labelling with Edu/Brdu. In this latter case, the base analogues were incorporated into each cell line prior to mixing them, i.e., when they were still growing in monoculture so they could be labelled and identified after co-seeding and morphological profiling. Each ground truth image was imported alongside the corresponding cell mask for that image. For each cell label, the fluorescence intensity was determined and tabulated. The threshold was set manually based on the fluorescence intensity of the monoculture controls. Ground truth labels were assigned to each region of interest (ROI).

### Feature extraction

Handcrafted features were extracted using the scikit-image package (regionprops and GLCM functions). The definition of each feature extracted from the image is listed in *Table 1*. All features were extracted for each channel in the cell painting image and for every region (cell, cytoplasm, and nucleus) within the ROI.

## Data filtering

Over the entire pipeline, ROIs could be discarded based on 3 conditions: (1) Cell detachment. Due to the harsh sample preparation and repeated washing steps, cells could detach from the imaging substrate thus resulting in lack of ground truth information for those cells. As a result, all ROIs for which there was no DAPI signal detected in the ground truth image, were removed from the dataset as incomplete. (2) Cells for which the ground truth fluorescence intensity was ambiguous. Ground truth labels were determined based on the presence of specific IF staining in either of the phenotype classes. If no class-specific IF staining was detected by thresholding, no true label could be assigned. These ROIs were, therefore, discarded due to uncertainty. (3) Likelihood of faulty ROI detection. The DAPI signal was used to discard ROIs that do not represent (complete) cells by thresholding on minimal DAPI intensity (mean nuclear DAPI intensity >500) and minimal nuclear size (nuclear area >160).

## ROI classification

### Train-validation-test split

Model training requires splitting the available data in three subsets: training (60%), validation (10%), and testing (30%). The training dataset was used to train the machine learning models (either RF or CNN). The validation dataset, composing of 10% of the total data, was used to determine the hyperparameters and intermediate testing. The remaining 30% of the dataset was kept apart and used to test the final model when training is completed. For both RF and CNN, the testing dataset was never shown to the model during the training phase, but only used after training to determine the accuracy of predictions to the ground truth. The number of instances for each class was equalized by sampling equal number of instances from each predicted class. To account for variation between technical replicates, the train-validation-test split was stratified per well. As a result, no datapoints arising from the same well could appear simultaneously in the training, validation and testing subset. This data stratification was repeated 3 times with different random seeds (see Methods – Reproducibility).

### Random forest

For each ROI, a set of manually defined parameters was extracted corresponding to cell shape (area, convex area, filled area, length minor axis, length major axis, centroid, eccentricity, equivalent diameter area, ferret diameter max, orientation, perimeter, solidity), texture (GLCM: contrast, dissimilarity, homogeneity, energy, correlation, ASM) and intensity (maximal intensity, minimal intensity, mean intensity, standard deviation intensity). This was done for three regions (nucleus, cytoplasm and complete cell), and for all channel dimensions. Redundant parameters could be removed if above a 0.95 correlation threshold. All parameters were standardized per ROI, grouped per replicate. The number of trees within the forest was varied between 10 and 150, reaching maximum accuracy at around 30 trees.

### Uniform manifold approximation and projection

Dimensionality reduction using UMAP was performed using either the same feature matrix as used for RF prediction or the feature embeddings from the trained CNN classification network. Hyperparameters were set at the default settings.

### Convolutional neural network

A ResNet50 model was trained for image classification. In contrast to classical machine learning techniques, no handcrafted features were extracted. Crops were defined based on the segmentation mask for each ROI. The bounding box was cropped out of the original image with a fixed patch size (60 µm for whole cells, 18 µm for nucleus and nucleocentric crops) surrounding the centroid of the segmentation mask. For the whole cell and nuclear crops, all pixels outside of the segmentation mask were set to zero. This was not the case for the nucleocentric crops. Each ROI was cropped out of the original morphological image and associated with metadata corresponding to its ground truth label. Images alongside their labels were fed to the network. Tensors were normalized per channel. Models are trained on a minimum of 5000 training inputs of each class for 50 epochs (training iterations). Each batch consisted of 100 samples. The training input was augmented by linear transformations (horizontal and vertical flip, random rotation). Each epoch, the current model was tested against a validation dataset (*Figure 5—figure supplement 2C*). The performance of the model on this validation

subset determined whether the model was stored as new best (if the new accuracy exceeded the accuracy of the previous best model) or discarded. The learning rate at the start was set at 0,0001 and automatically reduced with a factor of 0,1 during training when no improvement was seen after 10 epochs. After 50 epochs, the best resulting model was tested on a separate test dataset to determine the accuracy on previously unseen data.

### GradCAM

GradCAM analysis was used to visualize the regions used by the CNN for classification. This map is specific to each cell. Images are selected randomly out the full dataset for visualisation. To avoid cherry-picking, a set of GradCam maps is reported alongside the random seed used for image selection.

### Reproducibility

Each model training was performed three independent times (model initialisations, repetitions are indicated with 'N' on the figures). This was repeated for three different random seeds. Each model received input data arising from a minimum of 16 images per well, at least 15 technical replicates (wells). The optimisation experiments (*Figures 1–4*) were performed with cell lines with limited vari-ability. These models were trained on 3 independent experiments where ground truth pre-labelling (Edu/BrdU) was performed at least once on either of the cell lines in coculture. For iPSC-derived cultures, as variability is inherent to these differentiations, three biological replicates (independent differentiations) were pooled for model training.

### Statistics

All statistical comparisons were made nonparametric using Mann-Whitney U (for two independent sample comparison) or Kruskal-Wallis (for multiple sample comparison) with pairwise tests using Tukey's honestly significant difference test. We opted for nonparametric testing because the number of models in each group to be compared was <15. Significance levels are indicated on the figures using ns. (no statistical significance, p-value above 0.05), *(p-value between 0.05 and 5e-4), **(p-value between 5e-4 and 5e-6), and ***(p-value smaller than 5e-6). Error bars in the figures show standard deviation.

## Acknowledgements

We thank Marlies Verschuuren and Hugo Steenberghen for their assistance and knowledge regarding the cell lines. By extension, we would like to thank all current and former members of the De Vos lab. This work was funded by Fonds Wetenschappelijk Onderzoek Vlaanderen (1SB7423N; 1274822 N; G033322N; I000123N; I003420N), UAntwerpen (41739, 44742, 48955), and VLAIO (HBC.2023.0155).

## Additional information

### Funding

| Funder | Grant reference number | Author |
|---|---|---|
| Fonds Wetenschappelijk Onderzoek | 1SB7423N | Sarah De Beuckeleer |
| Fonds Wetenschappelijk Onderzoek | 1274822N | Johanna Van Den Daele |
| Universiteit Antwerpen | 41739 | Peter Ponsaerts<br>Winnok H De Vos |
| Universiteit Antwerpen | 44742 | Winnok H De Vos |
| Universiteit Antwerpen | 48955 | Winnok H De Vos |
| Agentschap Innoveren en Ondernemen | HBC.2023.0155 | Tim Van De Looverbosch |

| Funder | Grant reference number | Author |
|---|---|---|
| Fonds Wetenschappelijk Onderzoek | I000123N | Winnok H De Vos |
| Fonds Wetenschappelijk Onderzoek | I003420N | Winnok H De Vos |
| Fonds Wetenschappelijk Onderzoek | G033322N | Winnok H De Vos |

The funders had no role in study design, data collection and interpretation, or the decision to submit the work for publication.

## Author contributions

Sarah De Beuckeleer, Conceptualization, Formal analysis, Methodology, Writing – original draft; Tim Van De Looverbosch, Software, Formal analysis; Johanna Van Den Daele, Methodology; Peter Ponsaerts, Supervision, Writing – review and editing; Winnok H De Vos, Conceptualization, Supervision, Funding acquisition, Writing – original draft, Writing – review and editing, Project administration

## Author ORCIDs

Sarah De Beuckeleer ⓘ https://orcid.org/0000-0002-8964-4480
Tim Van De Looverbosch ⓘ https://orcid.org/0000-0002-3065-1395
Johanna Van Den Daele ⓘ https://orcid.org/0000-0001-8879-4551
Peter Ponsaerts ⓘ https://orcid.org/0000-0002-1892-6499
Winnok H De Vos ⓘ https://orcid.org/0000-0003-0960-6781

Joint public review: https://doi.org/10.7554/eLife.95273.4.sa1
Author response https://doi.org/10.7554/eLife.95273.4.sa2

# Additional files

## Supplementary files

MDAR checklist

## Data availability

The authors report that the results of this study are available within the manuscript and supplementary materials. All image analysis scripts are open-source available on GitHub (https://github.com/DeVosLab/Nucleocentric-Profiling, copy archived at *De Beuckeleer, 2024*) alongside a test dataset.

The following dataset was generated:

| Author(s) | Year | Dataset title | Dataset URL | Database and Identifier |
|---|---|---|---|---|
| De Beuckeleer S | 2024 | Nucleocentric-Profiling | https://doi.org/10.6084/m9.figshare.27141441.v1 | figshare, 10.6084/m9.figshare.27141441.v1 |

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
