## [Editor Report · eLife Assessment]

This study presents an **important** application of high-content image-based morphological profiling to quantitatively and systematically characterize induced pluripotent stem cell-derived mixed neural cultures cell type compositions. **Exceptional** evidence through rigorous experimental and computational validations support new potential applications of this cheap and simple assay.

---

## [Referee Report · Joint public review]

Automatically identifying single cell types in heterogeneous mixed cell populations hold great promise to characterize mixed cell populations and to discover new rules of spatial organization and cell-cell communication. Although the current manuscript focuses on the application of quality control of iPSC cultures, the same approach can be extended to a wealth of other applications including in depth study of the spatial context. The simple and high-content assay democratizes use and enables adoption by other labs.

The authors also propose a new nucleocentric phenotyping pipeline, where a convolutional neural network is trained on the nucleus and some margins around it. This nucleocentric approach improves classification performance at high densities because nuclear segmentation is less prone to errors in dense cultures.

The manuscript is supported by comprehensive experimental and computational validations that raises the bar beyond the current state of the art in the field of high-content phenotyping and makes this manuscript especially compelling. These include (i) Explicitly assessing replication biases (batch effects); (ii) Direct comparison of feature-based (a la cell profiling) versus deep-learning-based classification (which is not trivial/obvious for the application of cell profiling); (iii) Systematic assessment of the contribution of each fluorescent channel; (iv) Evaluation of cell-density dependency; (v) explicit examination of mistakes in classification; (vi) Evaluating the performance of different spatial contexts around the cell / nucleus; (vii) generalization of models trained on cultures containing a single cell type (mono-cultures) to mixed co-cultures; (viii) application to multiple classification tasks.

---

## [Author Response]

The following is the authors’ response to the previous reviews.

**Public Review:**
Summary:The authors present a new application of the high-content image-based morphological profiling Cell Painting (CP) to single cell type classification in mixed heterogeneous induced pluripotent stem cell-derived mixed neural cultures. Machine learning models were trained to classify single cell types according to either "engineered" features derived from the image or from the raw CP multiplexed image. The authors systematically evaluated experimental (e.g., cell density, cell types, fluorescent channels) and computational (e.g., different models, different cell regions) parameters and convincingly demonstrated that focusing on the nucleus and its surroundings contain sufficient information for robust and accurate cell type classification. Models that were trained on mono-cultures (i.e., containing a single cell type) could generalize for cell type prediction in mixed co-cultures, and to describe intermediate states of the maturation process of iPSC-derived neural progenitors to differentiation neurons.Strengths:Automatically identifying single cell types in heterogeneous mixed cell populations hold great promise to characterize mixed cell populations and to discover new rules of spatial organization and cell-cell communication. Although the current manuscript focuses on the application of quality control of iPSC cultures, the same approach can be extended to a wealth of other applications including in depth study of the spatial context. The simple and high-content assay democratizes use and enables adoption by other labs.The manuscript is supported by comprehensive experimental and computational validations that raises the bar beyond the current state of the art in the field of highcontent phenotyping and makes this manuscript especially compelling. These include (i) Explicitly assessing replication biases (batch effects); (ii) Direct comparison of featurebased (a la cell profiling) versus deep-learning-based classification (which is not trivial/obvious for the application of cell profiling); (iii) Systematic assessment of the contribution of each fluorescent channel; (iv) Evaluation of cell-density dependency; (v) explicit examination of mistakes in classification; (vi) Evaluating the performance of different spatial contexts around the cell/nucleus; (vii) generalization of models trained on cultures containing a single cell type (mono-cultures) to mixed co-cultures; (viii) application to multiple classification tasks.Comments on latest version:I have consulted with Reviewer #3 and both of us were impressed by revised manuscript, especially by the clear and convincing evidence regarding the nucleocentric model use of the nuclear periphery and its benefit for the case of dense cultures. However, there are two issues that are incompletely addressed (see below). Until these are resolved, the "strength of evidence" was elevated to "compelling".First, the analysis of the patch size is not clearly indicating that the 12-18um range is a critical factor (Fig. 4E). On the contrary, the performance seems to be not very sensitive to the patch size, which is actually a desired property for a method. Still, Fig. 4B convincingly shows that the nucleocentric model is not sensitive to the culture density, while the other models are. Thus, the authors can adjust their text saying that the nucleocentric approach is not sensitive to the patch size and that the patch size is selected to capture the nucleus and some margins around it, making it less prone to segmentation errors in dense cultures.

We agree that there is a significant tolerance to different patch sizes, and have therefore reformulated the conclusion as suggested in the results and the discussion sections (page 10 and 16). As very large patch sizes (>40µm) do increase the variability of the predictions and the imbalance between recall and precision, we have left this observation in the results section, as it also motivates for using smaller patch sizes.

Second, the GitHub does not contain sufficient information to reproduce the analysis. Its current state is sparse with documentation that would make reproducing the work difficult. What versions of the software were used? Where should data be downloaded? The README contains references to many different argparse CLI arguments, but sparse details on what these arguments actually are, and which parameters the authors used to perform their analyses. Links to images are broken. Ideally, all of these details would be present, and the authors would include a step-by-step tutorial on how to reproduce their work. Fixing this will lead to an "exceptional" strength of evidence.

We have added additional information to the GitHub to increase the reproducibility of the analysis.

• The README now contains additional documentation and more extensive explanations. A flowchart has been added, making the dataflow and order of analyses more clear.

• The accompanying dataset is 20GB in size and can be downloaded as a .zip-file from https://figshare.com/articles/dataset/Nucleocentric-Profiling/27141441?file=49522557. This file contains 2x480 raw images and a layout file.

• The used software versions are included in the manuscript in table 4. To increase the reproducibility, a Conda environment file (.yaml) has been added to the GitHub. This can be installed and contains the correct package versions.

• The README now contains for each script and its arguments a short description on its meaning, on whether it is required or optional and its default setting.

• A step-by-step tutorial on the use of the test dataset has been included. This tutorial includes the arguments used to run the code from the command line terminal.

**Recommendations for the authors:**
There are no reference from the text to Fig. 2D and to Fig. 3C.

This has been changed. The text has been added to the manuscript at page 6 (fig. 2D) and the reference to Fig. 3C has been included at page 8.